# German Translation and Cross-Cultural Adaptation of the Limb Deformity-Scoliosis Research Society (LD-SRS) Questionnaire

**DOI:** 10.3390/healthcare10071299

**Published:** 2022-07-13

**Authors:** Carolin Sophie Brune, Gregor Toporowski, Jan Duedal Rölfing, Georg Gosheger, Jana Fresen, Adrien Frommer, Andrea Laufer, Robert Roedl, Bjoern Vogt

**Affiliations:** 1Pediatric Orthopedics, Deformity Reconstruction and Foot Surgery, Muenster University Hospital, Albert-Schweitzer-Campus 1, 48149 Muenster, Germany; carolinsophie.brune@ukmuenster.de (C.S.B.); gregor.toporowski@ukmuenster.de (G.T.); jan.rolfing@rm.dk (J.D.R.); jana.fresen@ukmuenster.de (J.F.); adrien.frommer@ukmuenster.de (A.F.); andrea.laufer@ukmuenster.de (A.L.); robert.roedl@ukmuenster.de (R.R.); 2Children’s Orthopaedics and Reconstruction, Aarhus University Hospital, Palle Juul-Jensens Boulevard 99, J801, 8200 Aarhus, Denmark; 3General Orthopedics and Tumor Orthopedics, Muenster University Hospital, Albert-Schweitzer-Campus 1, 48149 Muenster, Germany; georg.gosheger@ukmuenster.de

**Keywords:** patient-reported outcome measure, translation and cultural adaptation, health-related quality of life, limb deformity, limb length discrepancy, LD-SRS

## Abstract

Background: Patient-reported outcome measures are gaining increasing importance in clinical research and quality control. Clinical impairment through limb deformities can appear in various forms. This study aimed at translating and culturally adaptating the Limb Deformity-Scoliosis Research Society (LD-SRS) patient-reported outcome measure (PROM) into German by following the scientific rigor of the cross-cultural adaptation process as well as ensuring the reliability of the translated version. The LD-SRS is applicable in children and adults. Methods: The translation was performed in accordance with the creators of the LD-SRS following the Professional Society for Health Economics and Outcomes Research (ISPOR) guidelines for translation and cultural adaptation. Two forward translations were performed, and after a consensus meeting, a professional translator translated the PROM back to English. The creators reviewed the back translation of the preliminary German version. Thirty patients with upper and lower limb deformities participated in cognitive debriefing interviews. The version was proofread and, finally, the test-retest reliability was estimated. Results: The mean age was 19 years (range 6–61). Twenty-six patients (87%) completed the retest after 6 days (range 3–26). The internal consistency was estimated with a Cronbach’s alpha of 0.96 (range 0.94–0.97), and the intraclass correlation was 0.92 (range 0.89–0.94), indicating an excellent reliability. The scores were normally distributed. Thereafter, the German version was proofread and finalized. Conclusions: The German translation and cross-cultural adaptation of the LD-SRS score resulted in a high reliability and internal consistency. The German LD-SRS score is readily usable and may be applied in future studies of German-speaking limb deformity patients.

## 1. Introduction

Limb deformities (LD) include the shortening or angular/rotational malalignment of the upper or lower extremities. LD are often associated with functional impairment of the affected extremity [1]. LD can arise from a variety of medical conditions (congenital or acquired due to trauma, tumor, infection as well as iatrogenic and idiopathic origin) [2,3]. Limb reconstruction is the art of treating LD and improving the functionality of the limb no matter the available resources [4].

Patient-reported outcome measures (PROMs) aim at identifying the health-related quality of life, mental health including self-image as well as symptoms, and functional status. Validated and effective outcome measures contribute to patient-centered care and thus to improving the treatment of the individual patient [5,6]. In the last decades, the use of PROMs has increased significantly [7,8]. Most PROMs focus on patients’ health status without including objective factors such as imaging, laboratory values, or clinician assessments [9]. A recent study showed that the perception of mental health, pain, or perceived function does not necessarily correlate with the severity of the deformity and is therefore not readily assessable by radiographic or objective clinical findings [1]. Furthermore, the self-perception of deformity, e.g., scoliosis, may not be correlated with the extent of the objectively measured deformity [10]. Over the past few decades, clinicians have recognized that patients’ perceptions are of utmost importance and are indispensable for clinical research and quality control in order to access and improve clinical practice [11]. 

PROMs may be conducted for assessing the general health status, e.g., the European Quality of Life 5 Dimensions (EQ5D) and the Short Form Health Survey (SF-36) are widely used [12]. Disease-specific PROMs for LD are scarce, but several research groups aim to establish and validate LD-PROMs [13,14,15,16,17,18].

While some PROMs are rather comprehensive and cover multiple domains, others are short and derived from similar existing disease-specific PROMs. In 1999, the Scoliosis Research Society (SRS) developed a patient-reported outcome measure (PROM) for patients with spine deformities in order to assess their quality of life. It consists of 30 questions covering the following five domains: function/activity, pain, self image/appearance, mental health, and satisfaction with the management/treatment [19]. During the development of the SRS questionnaire, five questions of the SF-36 were adaptated. Since there are many similarities between spine and limb deformity concerning pain, function and activity, and self-image, the questionnaire was modified by replacing the expression “back” or “trunk” with “limb”. This process was conducted by the Hospital for Special Surgery in New York and led to the Limb-Deformity SRS questionnaire (LD-SRS). The validation of the original English version of LD-SRS has been conducted, and the results showed that the score did not depend on demographic factors such as age, gender, or Body Mass Index (BMI) but did differ depending on whether treatment was imminent or had already been conducted [13]. Contrary to the SRS, the LD-SRS score has so far only been tested with patients older than 18 years [20,21,22]. 

To our knowledge, there is no PROM available that displays the impact of limb deformities in German. The aim of the present study was to translate and culturally adaptate the English version of the LD-SRS into German.

## 2. Materials and Methods

### 2.1. Translation Process

Creating different language versions of a PROM must not only be linguistically equivalent to the original version but also culturally adaptated. In some cultures, some idioms or expressions have either no meaning or a completely different meaning [7]. Following standardized guidelines with a multistep approach improves the translation quality significantly [23]. The World Health Organisation (WHO) and the Professional Society for Health Economics and Outcome Research (ISPOR) have developed guidelines for these processes. The WHO mentions five steps that must be followed, i.e., Forward Translation, Expert Panel, Back Translation, Pre-testing and Cognitive Interviewing, and the Final Version [24]. The ISPOR guidelines provide a more detailed framework, i.e., Preparation, Forward Translation, Reconciliation, Back Translation, Back Translation review, Harmonization, Cognitive Debriefing, Review of Cognitive Debriefing Results and Finalization, Proofreading, and Final Report [25].

In this study, the ISPOR guidelines were followed in order to create a German version of the LD-SRS:

Step 1: Preparation

The permission of the developer of the Original English Version of the LD-SRS was obtained, and ethical approval was applied from the local review committee: no. 2021-556-fS.

Step 2: Forward Translations and Reconciliation

Two independent German medical students performed the forward translation from English to German. Both translators were native German speakers and were fluent in English. None of the students knew that a German version of the related Scoliosis Research Society’s PROM for spinal deformities exists. Their two translations were compared with the German SRS-22. The translations of the individual items were close to the German version of SRS-22, and we therefore followed the English example by adaptating the German SRS-22, replacing the word “back” with “limb”. The consensus, a prelimary first German version, was reached through discussion and reconciliation.

Step 3: Backward Translation

A professional translator conducted a backward translation of the preliminary first German version. The translator had English as her mother tongue and was fluent in German. The translator was board-certified and a member in good standing of the German union of translators (https://bdue.de/der-beruf/beeidigte, accessed on 26 May 2022), which guarantees the highest professional standard. The back translation was reviewed and approved by the developers of the English LD-SRS. In the next step, the two German translators, the professional translator, and four German clinicians (JDR, GT, AF, BV) highlighted and investigated discrepancies between the original English version of the LD-SRS score and the back translation. Whenever discrepancies in terms of different meanings of the items occurred, the translation was revised. This process continued until the preliminary second German version was approved by all participants.

Step 4: Back Translation Review in Expert Panel Meeting

A meeting between the two independent German translators, the professional translator, and two German clinicians aimed to examine if all issues regarding the clinical perspective were measured and if it was understandable for patients. Then, this version was entered into REDCap (developed by Vanderbilt University, Nashville, TN, USA), an electronic data capture tool hosted at our university. Furthermore, scores ranging from 1 to 5 were assigned to response options according to the original LD-SRS score.

Step 5: Cognitive Debriefing Interviews

Participants with limb deformities were recruited in the outpatient clinic. Patients had to be fluent in German and able to give oral and written informed consent prior to participation in the study.

The reconciled German LD-SRS were emailed to the participants, and they were asked to fill them out to ascertain their individual comprehension. Three days later, the PROM was sent once more. The timing of the retest was decided on while considering on the one hand that the patients should not recall their first answer and on the other hand that the clinical status should not change in between the first and the second administration of the PROM. In limb lengthening and deformity correction, the clinical condition might change drastically within 1–2 weeks. 

Participants were requested to pay attention to the use of language and ambiguous expressions. The understanding, interpretation of different items, and cultural relevance were examined in the following interviews. The participants were asked if they needed the help of their guardian to understand or answer any question. It was proven that all instructions, questions, and response options were clear. All the interviews were conducted by one of the German forward translators. Potential problems with the translation, individual words, and items would have been discussed with the translators and clinicians. However, no ambiguity or problems were encountered in any age group.

Step 6: Proofreading and Finalization

Finally, the German clinicians proofread the tested German version of the LD-SRS, and the review did not reveal any grammatical or spelling mistakes. After this final approval, the German version was accomplished.

### 2.2. Statistical Analysis

The normal distribution was examined in all groups using the Shapiro–Wilk test. Descriptive statistics were performed using means and standard deviations or medians and ranges (minimum/maximum), depending on normal distribution testing for continuous variables. Cronbach’s alpha was used to examine the internal consistency, and, respectively, the intraclass correlation coefficient (ICC) was used to measure the test-retest reliability. Both values were reported while including the 95% confidence interval. Data were analyzed using SPSS Statistics v28.0.1.0 (IBM, SPSS, Inc., Chicago, IL, USA). 

## 3. Results

### 3.1. Study Participants

Thirty patients with a mean age of 19 years (range 6–61) participated in the cognitive interviews. According to Table 1, there were 13 children (twelve answered the retest), five teenager (three answering the retest), and twelve adults (eleven answering the retest). Additionally to gender and age, distinctions were made between the treated body parts (Table 1).

The median test-retest interval was 6.0 days (range 3–26). The wide range may be explained by the difficulty for participants and the caregivers to find time to schedule the cognitive debriefing interview, which took place after the retest. The results detected using the German LD-SRS are shown in Table 2. All items and domains were normally distributed.

### 3.2. Feasibility

No participant reported problems regarding the usability of the online version of the questionnaire. All patients felt that it was not time-consuming to answer the questionnaire, and some patients reported to have spent between 5 and 15 mins to answer the 30 items.

Few difficulties in understanding were named during the cognitive debriefing interviews in Step 5. Two patients did not know the German word “*Gliedmaße*” (“limb”) but could assume the correct meaning. Some questions, e.g., question 28 (“Has your treatment changed your confidence in personal relationships with others?”), were difficult to understand for young children, so they had to read the question twice. Since young children were allowed to make use of their parents’ help, some children did so.

Four children required help from a parent with question 15 (“Are you and/or your family experiencing financial difficulties because of your limb?”) because they did not know about the financial situation in their family. 

Regarding question 11 (“Which one of the following best describes your medication usage for your limb?”), two children did not know what kind of medicine they took and had to ask a parent. 

One young child could not remember when exactly the surgery took place, and three patients had several surgeries and, therefore, did not know which date should be given.

In general, both children and adults found the phrasing and wording of the 30 items easy to understand and did not mention major difficulties while filling them out. Based on the interviews, no items were changed in the final version.

### 3.3. Reliability and Internal Consistency

The statistical results are shown in Table 3. The evaluation of the internal consistency of the German version of the LD-SRS resulted in a Cronbach’s alpha of 0.96 in total. 

Concerning the domains, all Cronbach’s alpha values were ≥0.9. According to the literature, a Cronbach’s alpha can be interpreted as having a correlation that is excellent for >0.8 good for >0.8, acceptable for >0.7, questionable for >0.5 poor for >0.5, and unacceptable for <0.5 [26,27,28,29,30].

However, a wide range emerged when looking at the internal consistency of the individual items. For instance, questions 8 and 10 had a Cronbach’s alpha of 0.97 and 0.95, whereas question 28 resulted in a Cronbach’s alpha of 0.6. Only two items had a Cronbach’s alpha lower than 0.7. The overall reliability was thus excellent.

The ICC was 0.92 in total, which emphasizes an excellent reliability.

## 4. Discussion

The clinical constraints caused by limb deformities can manifest themselves in different ways. Several studies describe the mere acquisition of objective clinical data as insufficient, strengthening the need for PROMs that can be used to measure the needs and limitations of limb deformity patients. Accordingly, the original English version of the LD-SRS score was recently validated [13]. 

The aim of the present study to translate and culturally adaptate LD-SRS into German was achieved. The finalized LD-SRS–German version had an excellent test-retest reliability and is ready for clinical and scientific use. The rigorous ISPOR approach is the major strength of the present study. Following this guideline also helped other teams to translate and validate PROMs [31,32,33,34].

Most of the participants stated that the German LD-SRS questionnaire was easy to understand. However, some young children had problems understanding specific wording, i.e., “Gliedmaße”. Furthermore, small children sometimes did not know about the financial situation in their family or did not know what kind of medicine they took. Given their age and consequently their vocabulary and level of education, this seems reasonable. In general, the items were well understood, even by the youngest patients. Since the children were allowed to ask their parents questions while answering the questionnaire, patient age was not a limiting factor for the applicability of the LD-SRS score. 

The majority of the analysis showed a very good or excellent internal consistency. However, some questions, in particular question 4 (“If you had to spend the rest of your life with your limb shape as it is right now, how would you feel about it?”), question 26 (“Has your treatment changed your ability to enjoy sports/hobbies?”), and question 28 (“Has your treatment changed your confidence in personal relationships with others?”), scored the lowest Cronbach’s alpha, with values of 0.68, 0.67, and 0.6, respectively. In these cases, the median test-retest interval was detected as being longer than the corresponding subscale, so the physical conditions may have changed in the meantime. However, when analyzing the total German LD-SRS and its domains, an excellent reliability (Cronbach’s alpha > 0.9) was detected. The Cronbach’s alpha of the total LD-SRS score from the English version was comparable to our study, with 0.906.

During the translation process itself, some challenges appeared. As other studies pointed out [35], following every step provided by the guidelines was time-consuming, e.g., the recruitment of the back translator. Furthermore, finding patients willing to participate in the study in the midst of the COVID-19 pandemic was more challenging than anticipated. Another difficulty emerged when the retest had to be conducted. As the participation was based on voluntariness, patients often had to be reminded to fill out the retest by various emails or telephone calls, explaining the variation in the response time between the test and retest. Despite these challenges, the test-retest reliability was excellent, and only four patients did not complete the retest.

The reason to include mainly younger patients (<18 years) was that there are fewer comparable PROMs for children and that they might have more problems answering a questionnaire in English instead of their mother tongue. Furthermore, if young patients understand the wording and meaning of the items, older patients are inclined to understand these as well. The steps initiated in this study have the potential to be extended by applying the questionnaire to a more significant number of patients regardless of age, e.g., more adults undergoing limb deformity treatment and more patients with upper limb deformities. However, upper limb-specific PROMs, such as the Disability of the Arm, Shoulder, and Hand (DASH), Quick-DASH, or Patient Rated Wrist/Hand Evaluation (PRWHE), may be applied either in addition to or as an alternative to the LD-SRS score [36,37]. 

The study’s limitations include its sample size. Twenty-six patients completed the questionnaire twice. Furthermore, all the cognitive debriefing interviews were conducted by the same person. Here, problematic questions encountered during the first interviews may have been shifted into focus in the subsequent interviews. Compared to other emerging limb deformity PROMs, the LD-SRS score has the limitation of being adaptated from the spine deformity PROM SRS and consequently has not been designed explicitly for limb deformity patients. This may limit the content validity and the responsiveness of the questionnaire. Upcoming limb deformity-specific PROMs include the LIMB-Q Kids for younger patients, with a comprehensive approach to clarifying what matters to patients and ongoing multi-international field testing [14]. Moreover, the Stanmore Limb Reconstruction Score (SLRS) may also answer the call for a limb deformity-specific PROM [15,38,39]. In the future, the responsiveness and validity of these PROMS have to be investigated. It is likely that these PROMs may supplement each other and will be used in conjunction with each other, similar to the EQ5D and SF-36 for general health or the DASH and PRWHE PROMs for hand and wrist conditions. 

## 5. Conclusions

The German version of the LD-SRS was finalized and is ready for clinical application after a rigorous translation and cross-cultural adaptation process according to the ISPOR guidelines. The paper-based version of “LD-SRS German” can be downloaded from the Appendix A and the electronic version (REDCap codebook) can be requested from the authors. 

The test-retest reliability and internal consistency were very good/excellent. The authors also recommend the use of the German LD-SRS score in children with parental consultation. More evidence should be obtained in order to reliably assess the transferability of results to patients with upper limb deformities. 

## Figures and Tables

**Table 1 healthcare-10-01299-t001:** Demographics of patients participating in the cognitive debriefing interviews and the test-retest.

Variable	Test (*n* = 30/30)	Retest (*n* = 26/30)
Gender		
Male	15	12
Female	15	14
Age		
Child (6–13 y)	13	12
Teenager (14–17 y)	5	3
Adult (from 18 y)	12	11
Part of Body		
Thigh	14	11
Lower Leg	8	7
Foot/Ankle	6	6
Upper Arm	2	2
Forearm	0	0
Hand/Wrist	0	0
Total	30	26

**Table 2 healthcare-10-01299-t002:** Results of the German LD-SRS in total and its domains (SD = standard deviation).

Domain	Mean ± SD
Function/Activity	3.3 ± 1.0
Pain	3.2 ± 0.7
Self-Image	3.4 ± 0.6
Mental Health	3.5 ± 0.9
Satisfaction with Management	4.0 ± 0.6
Total	3.6 ± 0.9

**Table 3 healthcare-10-01299-t003:** Reliability and internal consistency of the German LD-SRS score and its subscales. Cronbach’s alpha and the intraclass correlation coefficient (ICC) were normally distributed; thus, both values are reported, including the 95% confidence interval.

Domain	Cronbach’s Alpha	ICC
Function/Activity	0.93 (0.82–0.98)	0.88 (0.69–0.95)
Pain	0.90 (0.74–0.96)	0.82 (0.59–0.93)
Self-Image	0.95 (0.87–0.98)	0.91 (0.78–0.96)
Mental Health	0.97 (0.92–0.99)	0.93 (0.85–0.97)
Satisfaction with Management	0.91 (0.77–0.97)	0.84 (0.62–0.93)
Total	0.96 (0.94–0.97)	0.92 (0.89–0.94)

## Data Availability

Not applicable.

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
