# Peer review of "German Translation and Cross-Cultural Adaptation of the Limb Deformity-Scoliosis Research Society (LD-SRS) Questionnaire"

_healthcare, 2022, doi:10.3390/healthcare10071299_

Round 1
Reviewer 1 Report
Dear authors,
congratulations for Your work that provides an important contribution in the dissemination of scoliosis research and prevention.
Sections are well structured, but some aspects in the discussion have to be pointed out. In particular, according to the available literature about scoliosis, the language of the questionnaires is not the only obstacle to overcome. So, You should briefly integrate the discussion deepening the other factors that usually could represent a limit in the scoliosis evaluation scale. In order to do that, I suggest the following references:
Notarnicola, A., Farì, G., Maccagnano, G., Riondino, A., Covelli, I., Bianchi, F. P., . . . Moretti, B. (2019). Teenagers’ perceptions of their scoliotic curves. an observational study of comparison between sports people and non- sports people. Muscles, Ligaments and Tendons Journal, 9(2), 225-235. doi:10.32098/mltj.02.2019.11
Scaturro D, de Sire A, Terrana P, Costantino C, Lauricella L, Sannasardo CE, Vitale F, Mauro GL. Adolescent idiopathic scoliosis screening: Could a school-based assessment protocol be useful for an early diagnosis? J Back Musculoskelet Rehabil. 2021;34(2):301-306. doi: 10.3233/BMR-200215. PMID: 33285626.
Best regards and good luck
Reviewer 2 Report
Dear Authors,
It is my pleasure to review your study. There is a lot of ambiguity in the articleI. I have a lot of doubts.
Introduction:
- in line 41-42: the use of PROMs has increased significantly [3].
Article number 3 is from 2015, it is not a good reference in this context.
-it is worth adding a hypothesis of your study.
-Limb Deformity should be better described.
M&M:
-how was the language proficiency of the translators (students) assessed?
Result:
-it is worth including (n =) in the Table No1
-how the children's responses were verified?
-how were the body parts explained / discussed to the respondents? patients often do not know anatomy
-in line 160 The median test-retest interval was 6.0 days (range 3-26). This is a wide range 3-26. Why?
- please explain the abbreviation in Table No 2
-children's self-responses / parental-assisted responses and teenagers' responses may have an impact on the outcomes and their reliability. The method of verification should be standardized. If someone does not report difficulties with the answer, it does not mean that his / her answer is reliable.
Discussion:
-the discussion could be better described with few relevant references.
Round 2
Reviewer 2 Report
Dear Authors,
In table 2 you can use the abbreviation SD and below the table explain it.
Currently, the article looks very good. A correction has been made. I have no objections. I think it can be published in Healthcare.